# A TriAdj-Adjuvanted *Chlamydia trachomatis* CPAF Protein Vaccine Is Highly Immunogenic in Pigs

**DOI:** 10.3390/vaccines12040423

**Published:** 2024-04-16

**Authors:** Jessica Proctor, Maria Stadler, Lizette M. Cortes, David Brodsky, Lydia Poisson, Volker Gerdts, Alex I. Smirnov, Tatyana I. Smirnova, Subarna Barua, Darren Leahy, Kenneth W. Beagley, Jonathan M. Harris, Toni Darville, Tobias Käser

**Affiliations:** 1Department of Population Health and Pathobiology, College of Veterinary Medicine, North Carolina State University, Raleigh, NC 27607, USA; 2Department of Biological Sciences and Pathobiology, Center of Pathobiology, Immunology, University of Veterinary Medicine Vienna, 1210 Vienna, Austria; 3Vaccine and Infectious Disease Organization (VIDO), University of Saskatchewan, Saskatoon, SK S7N 5A3, Canada; 4Department of Chemistry, North Carolina State University, Raleigh, NC 27607, USA; 5Department of Pathobiology, College of Veterinary Medicine, Auburn University, Auburn, AL 36849, USA; szb0116@auburn.edu; 6Centre for Immunology and Infection Control, Queensland University of Technology, Brisbane 4000, Australia; 7Department of Pediatrics, University of North Carolina, Chapel Hill, NC 27514, USA

**Keywords:** *Chlamydia trachomatis*, vaccination, adjuvant, chlamydial-protease-like activity factor, animal model, swine, one health

## Abstract

*Chlamydia trachomatis* (*Ct*) infections are the most common sexually transmitted infection (STI). Despite effective antibiotics for *Ct*, undetected infections or delayed treatment can lead to infertility, ectopic pregnancies, and chronic pelvic pain. Besides humans, chlamydia poses similar health challenges in animals such as *C. suis* (*Cs*) in pigs. Based on the similarities between humans and pigs, as well as their chlamydia species, we use pigs as a large biomedical animal model for chlamydia research. In this study, we used the pig model to develop a vaccine candidate against *Ct*. The vaccine candidate consists of TriAdj-adjuvanted chlamydial-protease-like activity factor (CPAF) protein. We tested two weekly administration options—twice intranasal (IN) followed by twice intramuscular (IM) and twice IM followed by twice IN. We assessed the humoral immune response in both serum using CPAF-specific IgG (including antibody avidity determination) and also in cervical and rectal swabs using CPAF-specific IgG and IgA ELISAs. The systemic T-cell response was analyzed following in vitro CPAF restimulation via IFN-γ and IL-17 ELISpots, as well as intracellular cytokine staining flow cytometry. Our data demonstrate that while the IN/IM vaccination mainly led to non-significant systemic immune responses, the vaccine candidate is highly immunogenic if administered IM/IN. This vaccination strategy induced high serum anti-CPAF IgG levels with strong avidity, as well as high IgA and IgG levels in vaginal and rectal swabs and in uterine horn flushes. In addition, this vaccination strategy prompted a pronounced cellular immune response. Besides inducing IL-17 production, the vaccine candidate induced a strong IFN-γ response with CD4 T cells. In IM/IN-vaccinated pigs, these cells also significantly downregulated their CCR7 expression, a sign of differentiation into peripheral-tissue-homing effector/memory cells. Conclusively, this study demonstrates the strong immunogenicity of the IM/IN-administered TriAdj-adjuvanted *Ct* CPAF vaccine candidate. Future studies will test the vaccine efficacy of this promising *Ct* vaccine candidate. In addition, this project demonstrates the suitability of the *Cs* pre-exposed outbred pig model for *Ct* vaccine development. Thereby, we aim to open the bottleneck of large animal models to facilitate the progression of *Ct* vaccine candidates into clinical trials.

## 1. Introduction

*Chlamydia trachomatis* (*Ct*) is a Gram-negative obligate intracellular bacterium with a biphasic developmental cycle (reviewed in [1]). While antibiotic treatment is available, *Ct* is with a global estimation of 129 million new *Ct* cases in 2020 [2] still the most prevalent bacterial sexually transmitted infection (STI). Difficulties with health care access and the lack of symptoms for most *Ct* infections have meant fewer patients have presented to the clinic for diagnosis and treatment. These untreated *Ct* infections can then cause chronic pelvic pain, ectopic pregnancies, and infertility in women [3,4,5]. Therefore, infection prevention is urgently needed. Currently, there are some *Ct* vaccine candidates in pre-clinical testing [6,7] and one candidate has completed clinical phase I [8]. So far, however, no vaccine is available to the public and additional research into *Ct* vaccine candidates is strongly needed.

Based on high similarities regarding size, physiology, reproductive cycles, immunology, and susceptibility to *Ct*, pigs have been used as biomedical animal models [9,10,11,12], including the study of immunity [13,14,15], STIs [16,17], and *Ct* vaccine development [18,19,20,21,22]. In addition, pigs can be infected both ocularly [23] and genitally [22,24] with *C. suis* (*Cs*). *Cs* is a chlamydia species not only closely related to *Ct*, but it is also associated with similar pathologies like conjunctivitis, respiratory infections, reproductive disorders, or enteritis [25,26]. *Cs* has furthermore been discussed to possess a zoonotic potential [27,28,29]. Previously, we have shown that pigs can be genitally infected with both *Cs* and *Ct*; the induced porcine CD4 T-cell response is cross-reactive between *Cs* and *Ct* [24]. This finding led us to use outbred *Cs* pre-exposed pigs as an animal model for *Ct* vaccine development. One advantage of using this pig model is that it closely mimics the clinical trial participant population that will likely be recruited for clinical (phase 3) trials—genetically diverse, high-risk *Ct* patients. High-risk *Ct* patients will have pre-existing immunity to *Ct*. Hence, our outbred pig model has a high chance of translating study outcomes, not only to a broader human population (including sexually active adults), but also to clinical trials. In a recent proof-of-principle vaccine study using TriAdj-adjuvanted UV-inactivated *Cs* as the vaccine, we demonstrated that this proof-of-principle vaccine induced a robust CD4 IFN-γ (=T-helper 1, Th1) response which is considered the mechanism of protection for various intracellular bacteria, including *Ct*, as reviewed in [5]. This Th1 response has been shown to be critical in the anti-*Ct* response [5,30,31,32,33]. In addition, upon challenge with *Cs*, our TriAdj-adjuvanted UV-*Cs* vaccine candidate reduced the genital *Cs* load compared to unvaccinated pigs, demonstrating vaccine efficacy. Thereby, we demonstrated that this *Cs* pre-exposed outbred pig model can be used to accurately assess both vaccine immunogenicity and efficacy.

After the successful establishment of this *Cs* pre-exposed outbred pig model, we now tested the immunogenicity of our first *Ct* vaccine candidate—TriAdj-adjuvanted chlamydial protease-like activity factor (CPAF). The TriAdj adjuvant consists of poly I:C, a host defense peptide, and polyphosphazene. It has been selected as an adjuvant for three main reasons: (i) it can be used intranasally and intramuscularly [34]; (ii) it induces effective humoral and cellular immunity with various vaccine antigens, including chlamydia [34,35,36]; and (iii) it strongly induces a Th1 response in our proof-of-principle *Cs* vaccination study [37].

As a vaccine antigen, CPAF was chosen based on its broad recognition by Th1 cells in human *Ct* patients. A previous study screened human *Ct* patients for *Ct* antigens that induced a CD4 T-cell-driven IFN-γ response. In this study, 16/30 patients elicited an IFN-γ response in PBMCs upon CPAF restimulation. With this response rate, CPAF was the most immuno-prevalent antigen in this study [38]. CPAF is a serine protease with no significant homology to other known genes [39]. CPAF is secreted into the host cell cytoplasm where it has a range of functions: (i) it can stabilize the intracellular inclusion in which *Ct* grows; (ii) it degrades pro-apoptotic BH3 to keep the infected host cell alive; (iii) it interferes with innate immunity, e.g., by blocking the activation of NF-κB or by interfering with NK cell activity by degrading CD1d; and (iv) it might even limit adaptive immunity by reducing MHC antigen presentation to CD4 and CD8 T cells [1].

After selecting the CPAF vaccine antigen, we selected the vaccination regimen. Based on the study of Abrahams et al. [8], we tested sequential intramuscular (IM) and intranasal (IN) vaccination regimens. Our two aims were as follows: (i) to determine the vaccine immunogenicity of the vaccine candidate and (ii) to determine if IM/IN vaccination or IN/IM vaccination is the more immunogenic vaccination regimen.

## 2. Materials and Methods

### 2.1. Animal Trial

The animal trial was designed, as illustrated in Figure 1. Eighteen 7-week-old pigs from the North Carolina State University (NC State) Swine Educational Unit were brought to the BSL-2 Laboratory Animal Research (LAR) facility at the NC State College of Veterinary Medicine (Raleigh, NC, USA). At this point, pigs had not developed anti-*Cs* antibodies (d.n.s.); however, as confirmed by *Cs*-specific qPCR [37], all pigs were qPCR-confirmed to be rectally infected with *Cs* (d.n.s.). Pigs were randomly distributed into three groups—MOCK, IN/IM, and IM/IN. After a resting period of three days (at 21 days post first vaccination, dpv), pigs were daily administered 1.44 g of doxycycline (Doxycycline Hyclate, West-Ward, Eatontown, NJ, USA) for four days and additionally 3 g of tylosin (Tylan soluble, Elanco^TM^, Indianapolis, IN, USA) twice a day for 3.5 days to treat pre-existing *Cs* infections. Then, a 14-day resting period was introduced to provide time for the anti-*Cs* immune response to subside. Thereafter, pigs received their first of four vaccinations at 7-day intervals according to their group allocation (0 dpv, see Figure 1 for group details). For simplicity reasons, the first two vaccinations were called “Prime” and the last two “Boost”. The IN/IM and IM/IN groups received TriAdj-adjuvant-formulated CPAF; MOCK groups received the TriAdj adjuvant without CPAF (for details, see below). Blood and rectal and vaginal swabs were taken prior to any vaccination day, as well as 6 days after the last vaccination, at −1, 6, 13, 20, and 27 dpv. One week after the last vaccination (28 dpv), pigs were sacrificed to collect uterine horn flushes. Blood was used to collect sera and PBMCs to study the systemic humoral and cellular anti-CPAF immune responses, respectively. Swabs and uterine horn flushes were used to analyze the mucosal anti-CPAF antibody response.

This animal trial was approved by the Institutional Review Board of the North Carolina State University (ID# 21-199B; approval date: 13 May 2021).

### 2.2. Vaccine Antigen Production and Formulation with TriAdj

CPAF was expressed in the *E.coli* strain BL21 DE3 using a codon-optimised open reading frame based on the *Ct* sequence WP_015506580 cloned into pRSETA. To facilitate expression, the first 26 residues of the open reading frame were omitted and proteolytic activity was compromised by including a S499→A499 substitution, as described by Chen et al. [40]. Open reading frames expressed from pRSETA were translated as fusion proteins that included the following: (i) an N-terminal poly-histidine tag enabling the immobilised metal affinity chromatograph, (ii) a T7 tag (Gene10 leader) which enhances foreign gene sequences in *E. coli*, and iii) an enterokinase cleavage sequence to allow the separation of N-terminal extensions from the open reading frame. The expressed protein was harvested in inclusion bodies which were washed extensively prior to urea solubilization and IMAC purification with Cobalt agarose (Talon^®^). Then, the urea was removed via dialysis against PBS, during which most of the expressed protein retained its solubility. Residual LPS was removed via cloud-point detergent extraction and the purified protein was lyophilised for storage. The TriAdj adjuvant was prepared according to the manufacturer’s instructions with the following final compositions per pig: 150 μg of poly I:C, 300 μg of a host defence peptide, and 150 μg of polyphosphazene (provided by the Vaccine and Infectious Disease Organization, VIDO, Saskatoon, SK, Canada). Within 1 h of vaccination, per pig, 30 μg of CPAF was formulated with the TriAdj adjuvant by simple mixing in the aqueous phase; afterwards, it was kept on ice until vaccine administration. For each vaccination site, the final vaccination volume was 1 mL per pig.

### 2.3. Isolation and Storage of Swabs, Sera, and PBMCs

For rectal swab collection, the swabs were inserted into the pig’s rectum. For vaginal swab collection, the vulva was cleaned, a speculum was inserted into the vagina, and the swab was rotated five times on the vaginal epithelium surrounding the cervix. Swabs were inserted into 1 mL of phosphate buffered saline (PBS) in a 1.5 mL snap-cap tube and stored on ice until further processing occurred in the laboratory. There, swab tubes were mixed using the vortexing method, and the swab was taken out of the liquid and rotated against the tube wall before being completely removed from the tube and discarded. The obtained swab samples were then frozen at −20 °C for future antibody quantification.

Blood was collected into SST and heparin vacutainers (BD Bioscience, San Jose, CA, USA) for serum and PBMC isolation, respectively. Blood was rested upright for at least 30 min before centrifugation was carried out at 2000× *g* for 20 min at room temperature (RT). Serum was harvested, aliquoted, and frozen at −20 °C for future anti-CPAF IgG analysis. The isolation of PBMCs was performed via density centrifugation using Sepmate tubes (StemCell, Vancouver, BC, Canada) and Ficoll-Paque (GE Healthcare, Uppsala, Sweden). After isolation, fresh PBMCs were used for in vitro restimulation to study the anti-CPAF-specific T-cell IFN-γ response via ELISpot and flow cytometry intracellular cytokine staining. Additionally, PBMCs were frozen in liquid nitrogen for future IL-17A ELISpots.

### 2.4. Interferon-γ and Interleukin-17A ELISpots

ELISpots were performed according to the manufacturer’s descriptions [MabTech, Nacka Strand, Sweden]. In brief, plates were activated with ethanol and coated overnight at 4 °C with either anti-IFN-γ [3130-3, MabTech] or anti-IL-17A [MT49A7 clone, MAbTech] capture antibodies. For IFN-γ ELISpots, fresh PBMCs, and IL-17A ELISpots, thawed PBMCs were seeded at 500,000 cells per well. Cells were stimulated for two days with 10 μg/mL of CPAF. Media and Concanavalin A (ConA, 5 μg/mL) were used as negative and positive controls, respectively. Afterwards, cells were washed and stained with the respective biotinylated detection antibodies, i.e., IFN-γ [3130-6, MabTech] or IL-17A [MTP853-biotin, MabTech]. Streptavidin–alkaline phosphatase combined with the 5-bromo-4-chloro-3-indolyl phosphate/nitro-blue tetrazolium substrate (100 μL/well, Sigma-Aldrich (St. Louis, MO, USA), Cat # B3679) was used to visualize spots. ELISpot plates were dried and spots were counted by a Mabtech ASTOR^TM^ ELISpot reader (Mabtech Inc., Cincinnati, OH, USA, for IFN-γ) or an AID ELISpot reader (AID, Straßberg, Germany) (for IL-17A). Data shown are based on three replicates.

### 2.5. Flow Cytometry (Including Intracellular Cytokine Staining)

Fresh PBMCs were plated in octuplicates at 500,000 cells/well in round-bottom 96-well plates and allowed to rest for 2–4 h. Afterwards, cells were stimulated overnight with 10 μg/mL of CPAF. Media and Concanavalin A (ConA, 5 μg/mL) were used as negative and positive controls, respectively. The following day, after 14 h of culture, Monensin (5 mg/mL, Alfa Aesar, Haverhill, MA, USA) was added for an additional 4 h to block Golgi transport. Replicates were pooled and stained for flow cytometry analysis according to Table 1. Data were acquired on a Cytoflex using CytExpert software (version 2.5, Beckman Coulter (Brea, CA, USA)). Data analysis was performed with FlowJo version 10.5.3 (FLOWJO LLC) with gates set based upon relevant FMO controls.

### 2.6. Anti-CPAF IgA, IgG, and IgM ELISAs (including IgG avidity)

For anti-CPAF IgA and IgG ELISAs, sera were diluted 1:5000 in an assay buffer (PBS +0.01% Tween-20 +0.1% bovine serum albumin, BSA); swabs and uterine horn flushes were used undiluted. Sample material to determine both IgA and IgG came from the same swabs. All assays were run in duplicates. First, polystyrene 96-well plates (NUNC MaxiSorp, Thermo Fisher Scientific, Waltham, MA, USA) were coated overnight at 4 °C with 10 μg/mL of CPAF. Plates were washed twice with wash buffer (PBS + 0.02% Tween 20) and blocked for at least 1 h at RT using 1% BSA in PBS. After four washes, diluted sera, swabs, and uterine horn flushes were added and incubated overnight at 4 °C. After four washes, horseradish peroxidase-conjugated anti-pig IgA or IgG detection antibodies were added to the wells (Bethyl Laboratories Inc., A100.104P and A100-117P, respectively) at a 1:200,000 dilution for 2 h at RT. After four final washes, substrate (3,3′,5,5′-Tetramethylbenzidine, TMB) was added and incubated for 30 min at RT. Colour reaction was quantified by optical density measurements at 450/620 nm using a Tecan Sunrise ELISA reader (Tecan, Männedorf, Switzerland).

For anti-CPAF serum IgG avidity assessment, a control or 6 M urea treatment was added after serum incubation. After washing, the wells were incubated for 10 min at RT either in the assay buffer (control treatment) or 6 M urea followed by four washes. The avidity index was calculated by dividing the OD value of the 6 M urea treatment by the OD value of the control treatment [41,42].

### 2.7. Statistical Analysis

Statistical analysis was performed using GraphPad Prism 10.2.0 (GraphPad Software, San Diego, CA, USA). Statistical significance was analyzed using either one-way (endpoint measurements) or two-way (longitudinal data) ANOVA, each with Dunnett’s multiple comparison test for within-group analyses and Tukey’s multiple comparison test for between-group comparisons. *p* < 0.05 was considered significant.

## 3. Results

The main goal of the study was to determine the vaccine immunogenicity of the TriAdj-adjuvanted *Ct* CPAF protein vaccine. Hence, prior to, during, and one week after the vaccination series, the humoral and cellular immune response was analyzed. Systemic cellular immunity was studied after the in vitro restimulation of PBMCs with CPAF via IFN-γ and IL-17A ELISpots (Figure 2) and via flow cytometric intracellular cytokine staining (Figure 3). The systemic humoral immune response was analyzed using serum anti-CPAF IgG ELISA, including antibody avidity testing (Figure 4). The mucosal antibody response was analyzed using anti-CPAF IgG and IgA ELISAs in vaginal and rectal swabs, as well as uterine horn flushes (Figure 5).

### 3.1. The Systemic Cellular Immune Response of IFN-γ and IL-17A ELISpots

CPAF-stimulated PBMCs from MOCK-vaccinated animals showed IFN-γ or IL-17A production at background levels (Figure 2, light blue, MOCK). In contrast, PBMCs from both CPAF-vaccinated groups increased both IFN-γ and IL-17A production over time (Figure 2, dark blue and purple groups). However, IN vaccination alone (blue group, 6 and 13 dpv) did not induce systemic IFN-γ production. In contrast, IM vaccination induced significant systemic IFN-γ production (6 and 13 dpv, purple group), which became significant as early as 13 dpv with IFN-γ spots of up to 800 spots per 500,000 PBMCs. While between-group comparisons were performed, they are, to avoid overloading the figures, not depicted in the graphs. IFN-γ production induced by IM/IN vaccination was also significant compared to MOCK (13 and 27 dpv) and IN/IM vaccination (13 dpv).

It must be noted that in contrast to the IFN-γ ELISpot which used fresh PBMCs, the IL-17A ELISpot was performed using thawed PBMCs that had been shipped from the original site of the study where the animal trial was performed (NCSU) to the current institution of Dr. Käser (the Vetmeduni Vienna). To ensure that only fully functional PBMCs were evaluated, a threshold of 200 IL-17A spots after ConA stimulation was applied. The consequent exclusion of several data points reduces the power of the IL-17A ELISpot assay. However, while the IL-17A data lack statistical significance, a similar trend is visible for IFN-γ: mainly the IM vaccination induced a systemic response with the highest values in the IM/IN vaccination group.

In summary, while both anti-CPAF vaccinations induced at least by number a systemic IFN-γ and IL-17A production, the IM/IN vaccination regimen induced strong and significant IFN-γ production.

**Figure 2 vaccines-12-00423-f002:**
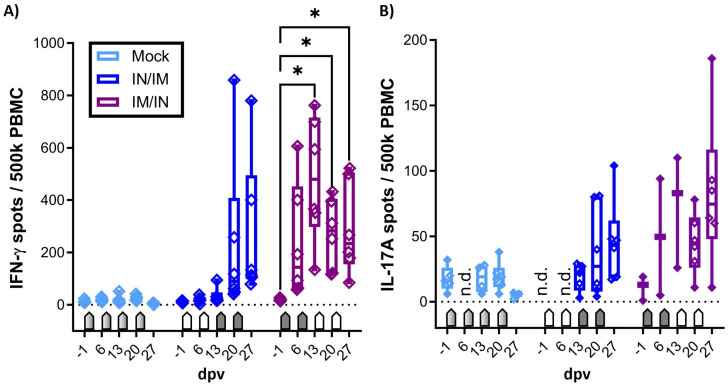
**Systemic IFN-γ and IL-17A production in response to in vitro CPAF restimulation.** (**A**) Systemic IFN-γ production was measured by ELISpot after the in vitro CPAF restimulation of fresh PBMCs. (**B**) IL-17A production was also measured by ELISpot upon in vitro CPAF restimulation using frozen and thawed PBMCs. The thawing process reduced the viability and responsiveness of some PBMCs. Only PBMCs with adequate response to control stimulation (ConA) were included in the results. Pentagonal arrows depict vaccinations: gradient: 50%IN/50%IM, white: IN, grey: IM. dpv = days after first vaccination. n.d. = not determined. The statistical analysis of within-group comparisons is shown. This statistical analysis was performed via GraphPad using 2-way ANOVA (with time and vaccination as the two parameters) and Dunnett’s multiple comparison test. * *p* < 0.05.

### 3.2. The Systemic Cellular Immune Response to Flow Cytometry

In addition to the IFN-γ and IL-17A ELISpots, the systemic cellular immune response of fresh PBMCs to in vitro CPAF restimulation was also analysed using flow cytometry. The staining panel was chosen not only to quantify IFN-γ production but also to identify the cellular source of IFN-γ, as well as homing patterns via the CCR7 expression of IFN-γ-producing cells (Table 1 and Figure 3A). As shown in the IFN-γ ELISpot, CPAF-stimulated CD4, CD8, and TCR-γδ T cells from MOCK-vaccinated animals showed IFN-γ production at low background levels (Figure 3B, light blue, MOCK). In this IFN-γ flow assay, PBMCs from the IN/IM CPAF-vaccinated groups also did not show increased IFN-γ production in either of the analysed cell subsets (Figure 3B, dark blue, IN/IM). In contrast, the IM/IN vaccination regimen induced significant IFN-γ production as early as 13 dpv. At this time point, the induced response was also statistically higher than in the MOCK and IN/IM groups (between-group statistical analysis is not shown). This systemic response was not further elevated by IN vaccination. The induced IFN-γ was solely produced by CD4 T cells (Figure 3B, dark blue, IM/IN).

The expression of CCR7 in T cells corresponds to lymph node homing, e.g., by naïve and central memory T cells. The loss of CCR7 expression is a sign of homing to non-lymphoid tissues, e.g., inflamed mucosal tissue [43]. Hence, CCR7 expression was analyzed in IFN-γ-producing CD4 T cells. Within the IFN-γ^+^ CD4 T cells, the IN/IM vaccination group only saw non-significant changes; in contrast, the IM/IN vaccination regimen led to an increased frequency of CCR7^−^ cells (Figure 3C).

In summary, IM/IN vaccination with TriAdj-adjuvanted CPAF induced significant IFN-γ production in CD4 T cells and induced their homing to non-lymphoid tissues, potentially including mucosal sites, like the genital or gastrointestinal tract.

**Figure 3 vaccines-12-00423-f003:**
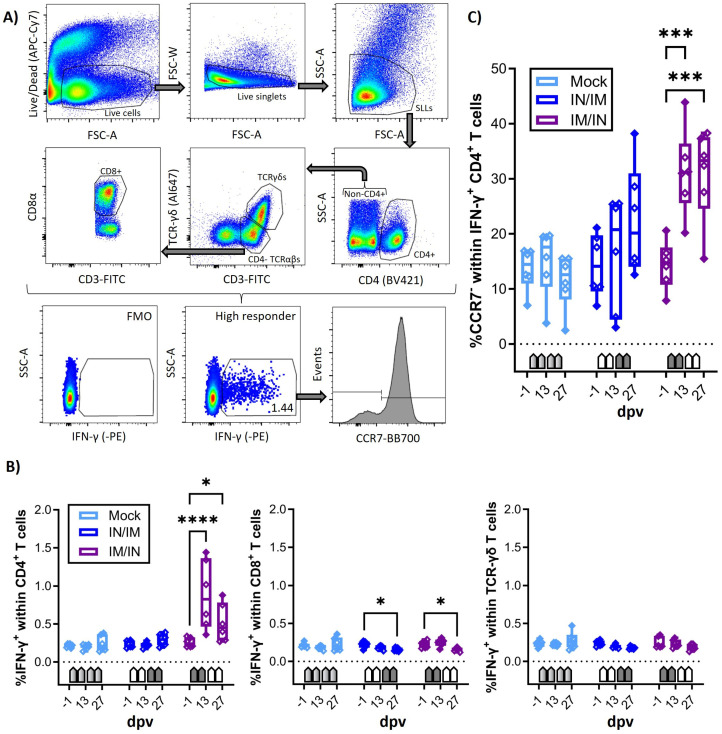
**IFN-γ production of T-cell subsets and differentiation of IFN-γ-producing CD4 T cells.** (**A**) shows the gating hierarchy to assess the IFN-γ response of T-cell subsets. A live/dead discrimination dye was included to exclude dead cells. Within live cells, doublets were excluded via a FSC-area (FSC-A)/FSC width (FSC-W) gate on live singlets. Live singlets were then used to identify live lymphocytes via a FSC/SSC single live lymphocyte (SLL) gate. These SLLs were used to gate on CD4 T cells (CD4/SSC-A), and CD4^−^ T cells were used to further discriminate TCR-αβ and TCR-γδ T cells (CD3/ TCR-γδ). CD4^−^ TCR-αβ T cells were further used to identify CD8 T cells (CD3/CD8α). Within CD4, CD8, and TCR-γδ T cells, IFN-γ production was analysed using intracellular cytokine staining (IFN-γ/SSC-A) with a gate set based on fluorescence minus one (FMO) control. Within IFN-γ-producing CD4 T cells, the expression of the lymph-node-homing marker CCR7 was analysed. (**B**) shows the frequency of IFN-γ^+^ cells within CD4 (left), CD8 (middle), and TCR-γδ (right) T cells after the in vitro CPAF restimulation of PBMCs from MOCK-vaccinated pigs (light blue), IN/IM-vaccinated pigs (dark blue), or IM/IN-vaccinated pigs (purple). Panel (**C**) shows the frequency of CCR7^−^ cells within IFN-γ-producing CD4 T cells. Pentagonal arrows depict vaccinations (gradient: 50%IN/50%IM, white: IN, grey: IM). Statistical analysis of within-group comparisons are shown using a 2-way ANOVA (with time and vaccination as the two parameters) and Dunnett’s multiple comparison test. **** *p* < 0.0001, *** *p* < 0.001, * *p* < 0.05.

### 3.3. The Systemic Humoral Immune Response: Anti-CPAF IgG Levels and Antibody Avidity

After demonstrating the induction of robust IFN-γ production with blood CD4 T cells, the systemic humoral response was analysed using serum anti-CPAF IgG ELISA (Figure 4A) and antibody avidity ELISA (Figure 4B). While serum IgG levels in MOCK animals stayed at background levels, anti-CPAF serum IgG levels in IN/IM-vaccinated pigs showed an increase at 27 dpv, but the varied response among animals prevented a statistically significant result. In IM/IN-vaccinated pigs, vaccination induced robust responses in all vaccinated pigs, resulting in a significant increase in serum anti-CPAF IgG levels at 20 and 27 dpv. (Figure 4A). This increase was also significant compared to the MOCK group (20 and 27 dpv) and the IN/IM group (20 dpv) (between-group statistical analysis not shown). Anti-CPAF IgG avidity was tested following treatment with 6M urea. Anti-CPAF IgG avidity was increased in sera from IN/IM-vaccinated animals at 27 dpv and for IM/IN-vaccinated animals as early as 13 dpv. While at 13 and 20 dpv, only the IM/IN group showed a significantly increased IgG avidity compared to both MOCK and IN/IM, at 27 dpv, both IN/IM and IM/IN vaccinations showed significantly increased avidity compared to MOCK. Conclusively, after the completion of the vaccination regimen, antibody avidity in both IN/IM- and IM/IN-vaccinated animals reached very high median avidity indices of >0.8. This shows that in anti-CPAF-vaccinated animals, over 80% of antibodies had a strong avidity to the *Ct* CPAF vaccine antigen (Figure 4B).

**Figure 4 vaccines-12-00423-f004:**
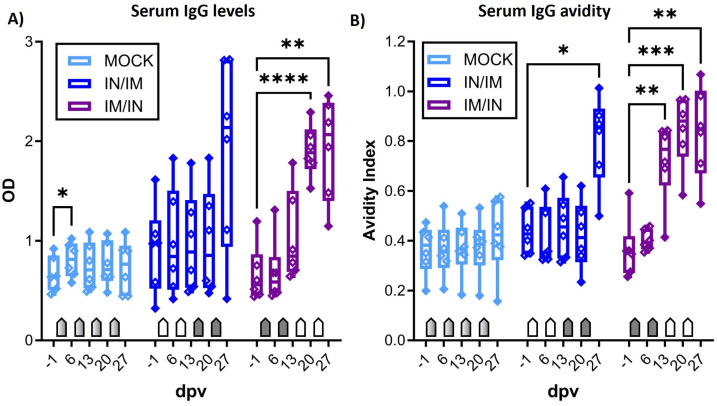
**Anti-CPAF IgG levels and avidity in sera.** (**A**) Immunoglobulin G (IgG) levels were quantified by anti-CPAF IgG ELISA in sera from animals receiving MOCK (light blue), IN/IM (dark blue), or IM/IN vaccination. Data show optical density (OD) values. (**B**) To assess IgG avidity in sera, a control or 6M urea treatment was included in the ELISA assay. Data show avidity index values which are calculated by dividing the OD value of the 6M urea-treated sample by the OD value of the control-treated sample. Pentagonal arrows depict vaccinations (gradient: 50%IN/50%IM, white: IN, grey: IM). Statistical analysis of within-group comparisons are shown. These statistical analyses were performed via GraphPad using 2-way ANOVA (with time and vaccination as the two parameters) and Dunnett’s multiple comparison test. **** *p* < 0.0001, *** *p* < 0.001, ** *p* < 0.01, * *p* < 0.05.

### 3.4. The Mucosal Humoral Immune Response: Anti-CPAF IgG and IgA Levels

The mucosal humoral immune response to CPAF was analysed via IgG and IgA ELISA in vaginal and rectal swabs, as well as in uterine horn flushes (Figure 5). In vaginal swabs, both TriAdj-adjuvanted CPAF vaccinations induced significant increases in anti-CPAF IgG levels (Figure 5A). For the IM/IN vaccination group, this increase was also statistically significant compared to the MOCK group. Vaginal anti-CPAF IgA levels were detected at lower levels and only IM/IN vaccination induced a significant increase at 13 dpv (Figure 5B). At 20 dpv, the IM/IN group had statistically significant higher IgA levels in vaginal swabs than the IN/IM group. Rectal swabs not only showed increased background levels but also higher data variability, especially for IgA (27 dpv data are not available for MOCK). Nevertheless, in rectal swabs, IM/IN vaccination once more led to an increased immune response with significantly higher anti-CPAF IgG and IgA levels after completing the IM/IN vaccination regimen (Figure 5C,D, 27 dpv). The increased IgG levels were statistically higher than in the IN/IM vaccination group. Anti-CPAF IgG and IgA levels were also quantified in uterine horn flushes to determine the humoral immune response in the upper genital tract. Both vaccination regimens led to increased anti-CPAF IgG and IgA levels in three (IgA, IN/IM) or four out of six vaccinated animals. In the IM/IN vaccination group, the increase in uterine horn anti-CPAF IgG levels was significant.

In summary, while both vaccination regimens led to increased mucosal IgG and/or IgA levels, at least by number, the IM/IN vaccination regimen produced more consistent responses, resulting in significantly increased anti-CPAF IgG and IgA levels at all analysed mucosal sites—in the gut, the lower genital tract, and the upper genital tract of most animals).

Taken together, IM/IN vaccination with the TriAdj-adjuvanted *Ct* CPAF vaccine candidate not only induced strong and significant systemic IFN-γ production by CD4 T cells with increased homing potential to non-lymphoid tissue, but also generated high serum levels of high-avidity anti-CPAF IgG and a robust mucosal anti-CPAF IgG and IgA response in the gut, as well as the lower and upper genital tract.

**Figure 5 vaccines-12-00423-f005:**
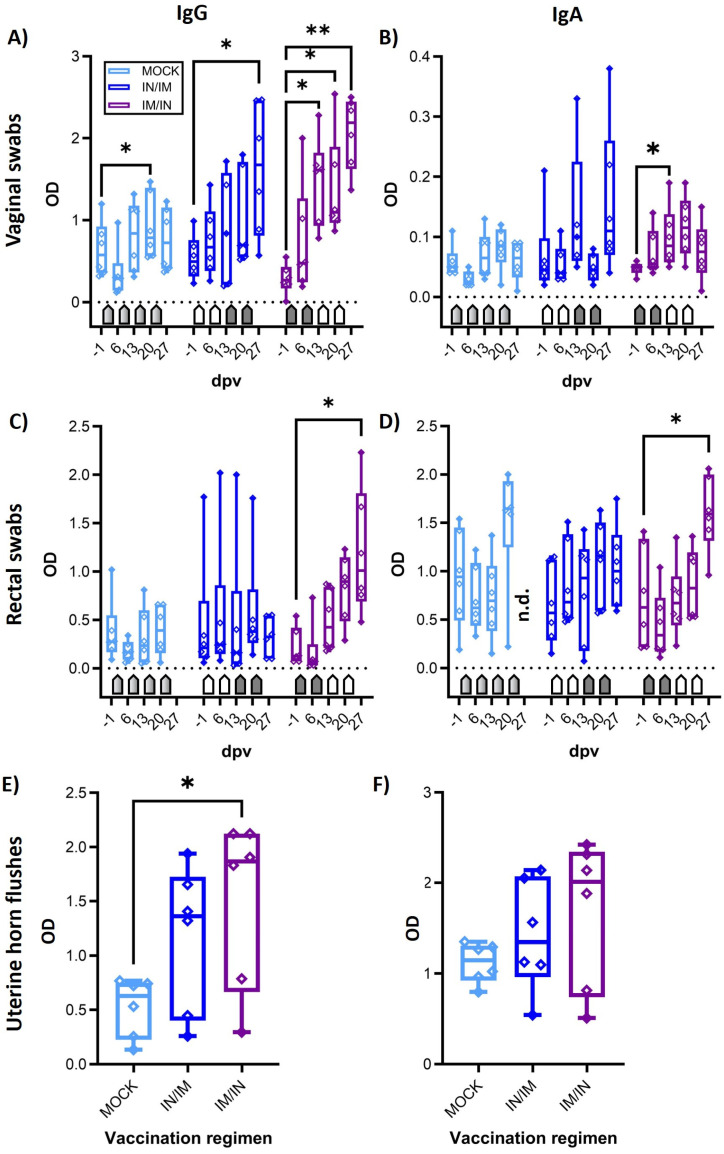
**Anti-CPAF IgG and IgA levels in vaginal and rectal swabs and in uterine horn flushes.** Immunoglobulin G levels (IgG (**A**,**C**,**E**) or IgA levels (**B**,**D**,**F**)) were quantified by isotype-specific anti-CPAF IgG or IgA ELISA in vaginal swabs (**A**,**B**), rectal swabs (**C**,**D**), or uterine horn flushes (**E**,**F**) from pigs receiving MOCK (light blue), IN/IM (dark blue), or IM/IN (purple) vaccination. Data show optical density (OD) values. Pentagonal arrows depict vaccinations (gradient: 50%IN/50%IM, white: IN, grey: IM). Statistical analysis of within-group comparisons are shown. This statistical analysis was performed via GraphPad using 2-way ANOVA (with time and vaccination as the two parameters) and Dunnett’s multiple comparison test. ** *p* < 0.01, * *p* < 0.05.

## 4. Discussion

Over the last ten years, we established *Cs*-pre-exposed outbred pigs as an animal model with an arguably high chance of translating data into human clinical trials [24,37]. Hence, the main goal of this study was to determine the vaccine immunogenicity of a TriAdj-adjuvanted CPAF vaccine candidate. While combinations of IM and IN administrations have been successful vaccination strategies, including for *Ct* vaccines [8], data are sparse on the optimal order of this combination for *Ct*–IN/IM or IM/IN. Hence, we compared the vaccine immunogenicity of our TriAdj-adjuvanted CPAF vaccine candidate in both orders.

The systemic response to IN/IM vaccine administration was often non-significant. The IN administration, in particular, did not induce much of a systemic immune response (both T-cell and antibody responses) at the earlier time points (6 and 13 dpv). This is in line with our previous data in which IN vaccination using UV-inactivated *Cs* adjuvanted by TriAdj failed to induce a systemic IFN-γ response pre-challenge. In this previous study, however, vaccination primed CD4 T cells for a stronger post-challenge IFN-γ response and induced their differentiation into effector-memory T cells. This priming of CD4 T cells can explain the faster and stronger immune response in the genital tract and the observed decrease in *Cs* particles in the vaccinated animals [37]. The most recent data on *Ct* vaccination in a clinical phase I trial also lacked a further increase in IM-induced systemic immunity levels caused by IN boosts [8]. Nevertheless, one should not underestimate the benefits of IN vaccination without a detailed analysis of the mucosal immune response. Stary et al. also demonstrated that IN vaccination can induce potent protection by elevating mucosal immunity. In that study, the systemic response was also minimal, as the only induction of systemic immunity data was a transient and non-significant increase in mostly *Ct*-specific NR1 cells. This Stary et al. study further demonstrates that, even in the absence of the major induction of systemic immune responses, IN vaccination can be highly protective [44]. Hence, future studies on *Ct* vaccination should include a comprehensive analysis of mucosal immunity to optimally demonstrate vaccine immunogenicity, especially for IN vaccination.

In contrast to the minimal effects of IN vaccination on systemic immunity, our results demonstrate that in the IM/IN vaccination regimen, the TriAdj-adjuvanted CPAF vaccine candidate induces a strong systemic Th1 response and the differentiation of IFN-γ-producing CD4 T cells into CCR7^−^ cells. It is well established that this Th1 response is crucial in offering protection against *Ct* [5]. Furthermore, CCR7^−^ Th1 cells have the ability to home to mucosal tissue; these mucosal-tissue-homing cells have been further implicated in protection against *Ct* [44]. Therefore, the IM/IN administration of the TriAdj-adjuvanted CPAF vaccine candidate induces a cellular immune response that has been most strongly linked with protection against *Ct*.

In addition to the Th1 response, the humoral response contributes to protection against genital tract *Ct* reinfections [45,46]. The humoral immune response seems to mainly contribute to antibody-induced cellular responses through Fcγ receptors. Moore et al. named enhanced phagocyte-induced *Ct* killing through increased opsonophagocytosis, the FcR-mediated enhancement of antigen presentation, the inhibition of *Ct* growth, and antibody-dependent cellular cytotoxicity (ADCC) as examples; they further suggest that “a future anti-chlamydia vaccine should elicit both humoral and T-cell-mediated immune responses for optimal memory response and vaccine efficacy” [45]. However, these benefits of antibodies may only apply when the immunogen is on the outer membrane of *Ct*. CPAF is a chlamydial protein that is secreted into the host cytosol and released into the extracellular milieu when the host epithelial cell lyses. It was reported that CPAF can limit neutrophil recruitment [47], degrade host antimicrobial peptides [48], paralyze neutrophils, and inhibit their chlamydiacidal activities [49]. It is possible that antibodies could neutralize this function of CPAF. Besides inducing a strong Th1 response, we show that IM/IN vaccination induced a pronounced humoral immune response: the systemic anti-CPAF IgG response was strongly increased and antibody avidity was significantly elevated. Besides being an indication for successful germinal center formation and antibody maturation, antibody avidity can correlate well with cross-neutralization, e.g., for SARS-CoV-2 [41]. In addition, Stanley Plotkin concluded (at least for *Haemophilus influenzae* type b) that “antibodies must be of high avidity in order to protect” [50]. Therefore, the induction of high levels of high-avidity serum anti-CPAF IgG antibodies demonstrates a potent induction of a relevant systemic immune response.

Besides the systemic immune response, anti-CPAF IgG and IgA levels have been analyzed in mucosal samples, such as vaginal swabs, rectal swabs, and uterine horn flushes. As systemically shown, the IM/IN administration of TriAdj-adjuvanted CPAF induced a strong antibody response with increased IgA levels, and particularly high and significantly increased IgG levels at basically all analyzed mucosal sites (rectally and in the lower and upper female genital tract). These high mucosal anti-CPAF antibody levels indicate that the IM/IN administration of the TriAdj-adjuvanted CPAF vaccine candidate induces a prominent local humoral immune response. Together with the systemic Th1 response and high-avidity IgG antibodies, these mucosal antibodies have the potential to limit genital and rectal *Ct* infections.

## 5. Conclusions

In conclusion, in our *Cs* pre-exposed outbred pig model, the IM/IN administration of the TriAdj-adjuvanted CPAF vaccine candidate not only induces a strong systemic Th1 and IgG response but also high mucosal antibody levels against the broadly recognized *Ct* CPAF protein. The strong immunogenicity in a relevant large biomedical animal model indicates that TriAdj-adjuvanted CPAF is a viable vaccine candidate against *Ct*. Future studies should extend the vaccine immunogenicity analysis to also determine vaccine efficacy. These vaccine efficacy data will then be able to demonstrate if the TriAdj-adjuvanted CPAF vaccine candidate can limit *Ct* infections (the most prevalent bacterial STI).

## Figures and Tables

**Figure 1 vaccines-12-00423-f001:**
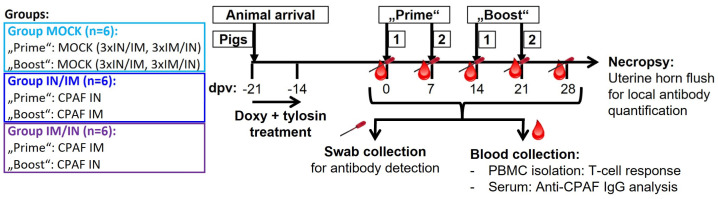
**TriAdj-adjuvanted** ***Ct*** **CPAF vaccination trial (groups and layout).** Eighteen 7-week-old female pigs were distributed into three groups—MOCK, intranasal/intramuscular (IN/IM), and IM/IN vaccination. After a one-week antibiotic treatment and two resting weeks to allow the anti-*Cs* response to decline, pigs were vaccinated according to their group allocation at 0, 7, 14, and 21 days post (first) vaccination (dpv). As indicated in the timeline, blood and vaginal/rectal swabs were collected throughout the study the day before each vaccination and at six days after the last vaccination (27 dpv) to assess the humoral and T-cell immune response. The day after the last blood collection (28 dpv), pigs were sacrificed to collect uterine horns for local antibody quantification.

**Table 1 vaccines-12-00423-t001:** Flow cytometry staining panel.

Antigen	Clone	Isotype	Fluorochrome	Labelling Strategy	Primary Ab Source	2nd Ab Source
CD3	PPT3	IgG1	FITC	Direct conjugation	Southern Biotech	-
CD4	74-12-4	IgG2b	Brilliant Violet 421	Secondary antibody	BEI Resources	Jackson Immunoresearch
CD8α	76-2-11	IgG2a	Brilliant Violet 605	Biotin–streptavidin	Southern Biotech	Biolegend
TCR-γδ	PGBL22A	IgG1	Alexa Fuor 647	In-house conjugation ^1^	Invitrogen	-
CCR7	3D12	rIgG2a	Brilliant Blue 700	Direct conjugation	BD Biosciences	-
Live/Dead	-	-	Near-infrared	-	Invitrogen	-
IFN-γ	P2G10	IgG1	PE	Direct conjugation	BD Biosciences	-

^1^ In-house conjugation was performed using Invitrogen Alexa Fluor™ Antibody Labelling Kits (Thermo Fisher Scientific).

## Data Availability

The original contributions presented in this study are included in the article. Further inquiries can be directed to the corresponding author.

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
