# Peer review of "A TriAdj-Adjuvanted Chlamydia trachomatis CPAF Protein Vaccine Is Highly Immunogenic in Pigs"

_vaccines, 2024, doi:10.3390/vaccines12040423_

Round 1

Reviewer 1 Report

Comments and Suggestions for Authors

Chlamydia trachomatis (Ct) is a fascinating human pathogen causing a common bacterial infection in humans and accounts for significant morbidity worldwide. Ct infection can be asymptomatic and result in (i) the leading infectious cause of preventable blindness worldwide called trachoma and (ii) pelvic inflammatory disease, which can lead to infertility and ectopic pregnancy. Although intensive efforts to develop a trachoma vaccine, including human trials dating back to the 1960s, no vaccines for the disease are currently available primarily due to the Ct pathogen complexity and the lack of efficient adjuvants and animal models.

            Therefore, Proctor et al.'s study is important because it demonstrates that pre-exposed pigs could serve as a suitable animal model for preclinical testing of the safety and immunogenicity of vaccine candidates.

            There are a few issues I think it should be resolved before publication:

INTRODUCTION

-        Please add the advantages and disadvantages of the pig model to the Introduction: Are pigs used as experimental animals for genital and ocular infection? Are Ct and Cs experimental genital infections in pigs done in a similar manner? What are the dosages for both genital infections? Are there differences in symptoms and longevity of infection? It would be more interesting for the general readership.

METHODS

-        Please add information about TriAdj adjuvant here (or in the Intro or Discussion). Is it safe to use TriAdj on mucosal surfaces? What is its (TriAdj) composition? Please add the manufacturer.

-        Please write a more detailed protocol on how you produced CPAF. Was it the full-length protein? Have you had any issues with CPAF precipitation in a solution without urea? Please add these details.

-        Was the concentration of CPAF 300 mcg/animal? Or per ml? What was the final dose volume that the animals received? Please add these details.

-        Were there any differences between the IM and IN application formulations? Please add these details.

-        Lines 142-147: how did you ensure that IgG and IgA recovery from swabs was uniform?

RESULTS

Please clarify in the text (also in the Discussion):

-        Chlamydia-caused diseases are local diseases. Why would systemic-produced IFNg and IL-17A be of importance?

-        Did you expect an increase in systemic IFNg when immunizing IN in your system? This is not what usually occurs when other immunogens are used.

-        Lines 267-269. What is your evidence that CPAF-specific CD4+CCR7- cells arrived at other mucosal sites? Please reformulate.

-        Lines 267-269. The genital and GI tract have their corresponding lymphoid sites: gut-associated lymphoid tissue and vagina-associated lymphoid tissue. Please reformulate your sentence.

DISCUSSION

-        Line 347. Please update data about estimated Ct infections; there are more recent data than from 2016.

-        Line 353. Please describe in more detail and specify what you mean by long-term humoral and cellular (in years).

-        Lines 362-264. I would rewrite this because IN immunization does not generally induce a high systemic response.

-        Lines 400-401. I would tone down the statement that Chlamydia paralyzes neutrophils. Yes, a paper from Rudel’s group claims this, but there is also a lot of controversy about it in the field.

-        Lines 405-410. While some avidity antibodies may indeed be neutralizing antibodies, not all antibodies with high avidity are necessarily neutralizing. Avidity relates more to the strength of the interaction between an antibody (paratope) and its antigen (epitope). At the same time, neutralizing activity specifically refers to the ability of an antibody to neutralize the infectivity or pathogenicity of a microorganism. Moreover, the antibody avidity is not proof of successful germinal center formation. Please reformulate this part.

Best wishes,

Reviewer 2 Report

Comments and Suggestions for Authors

The manuscript from Proctor et al. investigates the immunogenicity of a TriAdj-adjuvanted C. trachomatis CPAF protein vaccine in pigs. The rationale for using a pig model is sound and supported by previous studies. Overall, this is a well-written study, with minor English corrections needed, and the data are generally sound and reproducible. I feel that the manuscript can be improved though, and mainly with a better structured and informed Introduction.

I would also contend that this is not an immunogenicity trial, as no live challenge has been done; it is strictly an antigenicity trial and the title should reflect this , e.g. Antigenicity of a Tri…CFAP (in full) protein vaccine in pigs.

 Introduction: in my opinion the introduction needs major revision. For the non-Chlamydiologist reader, there is insufficient information about the bacterium (e.g. intracellular pathogen lifestyle, RB/EBs, cell wall structure, etc) and nothing about global incidence to demonstrate that it’s the primary STI. Secondly, line 65-68, why is this population being considered? I assume they have had previous Ct infection (e.g. sex workers, etc), but why not consider a Ct vaccine for a broader population of sexually active adults (and introduce in the same way as the HPV vaccine)? This should be discussed.

Line 71: The mechanism of protection is a Th1 CD4 driven IFNy response, which is common for intracellular pathogens. This is not made clear in the introduction and needs careful explanation.

Line 78-81: there is absolutely little about the antigen chosen, other than how it was chosen. I would like to see the inclusion of some description of the biology, structure and function of CPAF, whether it has any homologues in other bacteria (is it unique to Ct?).

Lines 82-95 should be deleted. These are a summary of methods and results that are out of place in an introduction. I would instead provide a hypothesis or aims.

Materials and Methods:

Line 102: pigs didn’t show anti-Cs antibodies, but PCR picked up rectal infection with Cs, necessitating a course of antibiotics and rest period. The authors should confirm that post-treatment, there were no anti-Cs antibodies and PCR results now negative. This needs to be made clear that animals are naïve for immunization.

Line 113-114: swabs and bloods taken from which sites? Vaginal/rectal (noted in legend, move to main text as well)

Line 112/134. The TriAdj adjuvant – who makes it (manufacturer’s instructions?) and can authors provide any references for this. Actually, details of the adjuvant should go in the introduction as well, as many readers will be unfamiliar with this adjuvant.

Lines 129-138. Details are a bit vague on the CPAF. Is there a reference for this antigen that can be provided here?  How is antigen formulated with adjuvant? Is it simple mixing in aqueous phase? Or is it an emulsion?

Line 158. Manufacturer’s instructions  -I presume it is Mabtech?

Line 203: Statistics – add p value for significance ( p <?).

Results:

 Figure 4& 5. Is there any statistical difference between the IN/IM and IM/IN protocols for the generation of IgG, IgA and avidity? That is, is one protocol better than the other? Data suggest that it is IM/IN, but is this supported statistically?

Authors might  want to revisit their figures; the heavy coloured bars makes it difficult to see the individual points. I would change to thin lines at least.

Discussion:

1.       Reference [22] should be in introduction with general background.

2.       Line 351 - I would not call this an immunogenicity trial; it is an antigenicity trial, whereas a true immunogenicity trial would involve challenge of the animals with live Cs/Ct and then examine colonisation and the immune response.

3.       Some of the above-mentioned points from the introduction can be gleaned from the discussion (cut them and paste and expand).

Comments on the Quality of English Language

First sentence in the introduction does not scan. Should be broken into 2 sentences.

Line 144; vortexed, preferable to write mixed by vortex.

Immunogenicity should be replaced with antigenicity, unless you are describing a live bacterial challenge expt or study.
